# Neuroprotective Effect of 3′,4′-Dihydroxyphenylglycol in Type-1-like Diabetic Rats—Influence of the Hydroxytyrosol/3′,4′-dihydroxyphenylglycol Ratio

**DOI:** 10.3390/nu14061146

**Published:** 2022-03-08

**Authors:** María Dolores Rodríguez-Pérez, Inmaculada Pérez de Algaba, Esther Martín-Aurioles, María Monsalud Arrebola, Laura Ortega-Hombrados, Cristina Verdugo, María África Fernández-Prior, Alejandra Bermúdez-Oria, José Pedro De La Cruz, José Antonio González-Correa

**Affiliations:** 1Department of Pharmacology, Biomedical Research Institute (IBIMA), School of Medicine, University of Málaga, 29010 Málaga, Spain; loladoct@uma.es (M.D.R.-P.); lauraortegah@outlook.es (L.O.-H.); cristinaverdugocabello@uma.es (C.V.); correa@uma.es (J.A.G.-C.); 2Clinical Laboratory Department, Public Hospital, Montilla, 14550 Córdoba, Spain; iperezdealgaba@gmail.com; 3UGC La Roca, District Sanitary of Málaga-Guadalhorce, 29001 Málaga, Spain; esther.uma@hotmail.com; 4UGC Clínical Laboratory, Hospital Axarquía, AGSEMA, 29740 Málaga, Spain; mariam.arrebola.sspa@juntadeandalucia.es; 5Instituto de la Grasa, Consejo Superior de Investigaciones Científicas (CSIC), Ctra. Utrera Km 1, Campus Universitario Pablo de Olavide, Edificio 46, 41013 Seville, Spain; mafprior@ig.csic.es (M.Á.F.-P.); aleberori@ig.csic.es (A.B.-O.)

**Keywords:** hydroxytyrosol, 3′,4′-dihydroxyphenylglycol, extra virgin olive oil, diabetic neuropathy, diabetes mellitus

## Abstract

The aim of this study was to assess the possible neuroprotective effect of 3′,4′-dihydroxyphenylglycol (DHPG), a polyphenol from extra virgin olive oil (EVOO), in an experimental model of diabetes and whether this effect is modified by the presence of another EVOO polyphenol, hydroxytyrosol (HT). The neuroprotective effect was assessed in a hypoxia–reoxygenation model in brain slices and by quantifying retinal nerve cells. The animals were distributed as follows: (1) normoglycemic rats (NDR), (2) diabetic rats (DR), (3) DR treated with HT (5 mg/kg/day p.o.), (4) DR treated with DHPG (0.5 mg/kg/day), or (5) with 1 mg/kg/day, (6) DR treated with HT plus DHPG 0.5 mg/kg/day, or (7) HT plus 1 mg/kg/day p.o. DHPG. Diabetic animals presented higher levels of oxidative stress variables and lower numbers of neuronal cells in retinal tissue. The administration of DHPG or HT reduced most of the oxidative stress variables and brain lactate dehydrogenase efflux (LDH) as an indirect index of cellular death and also reduced the loss of retinal cells. The association of DHPG+HT in the same proportions, as found in EVOO, improved the neuroprotective and antioxidant effects of both polyphenols.

## 1. Introduction

In the evolution of diabetes mellitus, controlling blood glucose levels not only serves to improve the metabolic profiles of patients but also reduces the appearance and/or intensity of long-term cardiovascular complications [1]. Diabetic macroangiopathy does not differ from the arteriosclerosis process among the general population, although it occurs earlier in time and is more intense in its vascular alterations, especially those of an ischemic nature [2]. Diabetic microangiopathy is typical of diabetes mellitus and can appear in the kidney (diabetic nephropathy), eye (diabetic retinopathy), or nervous tissue (diabetic neuropathy) [3].

Diabetic nervous tissue damage can occur in the peripheral nerves or in the central nervous system [4,5]. A greater sensitivity of the central nervous tissue to ischemic damage, independent of diabetic vascular alterations, was previously described in models of experimental diabetes [6,7]. Likewise, neuronal damage even earlier than vascular damage was described in the retinas of diabetic animals [8,9].

In all these microangiopathic processes, oxidative stress is the key factor for the initiation of all other biochemical pathways that lead to tissue damage (nervous tissue, kidney, or retina) [10,11]. An increase in nicotinamide adenine dinucleotide oxidase (Nox) activity was described in endothelial and nerve cells in the early stages of diabetes mellitus and lead to the production of superoxide anions [12]. On the other hand, sustained hyperglycemia alters the intracellular metabolic pathways that control glucose metabolism, such as the polyol pathway, hexosamine pathway, advanced glycation end products (AGEs), and protein kinase C [13]. All these pathways are generators of free radicals, and, in turn, these radicals increase the activity of the aforementioned biochemical pathways. The production of superoxide and hydroxyl anions modifies various cellular chemical elements, leading to an excess of lipid peroxides, damage to the DNA structure, an increase in protein carbonyls, etc. [10]. Deficits in endogenous antioxidant defense mechanisms, such as superoxide dismutase, catalase, and the glutathione system, have also been described [14].

In relation to some complications of diabetes, increases in the production of lipid peroxides (tissue and serum), 8-isoprostanes, and markers of DNA damage, 3-nitrotyrosine (nitrosative stress biomarker), as well as decreases in essential components of the glutathione system and total antioxidant capacity in the tissues and sera of diabetic animals, were previous observed under the experimental model of diabetes mellitus used in this study [6,11,15].

For this reason, in addition to controlling blood glucose levels, it was postulated that a decrease in oxidative stress could help in the prevention of these alterations [16].

In this sense, one of the greatest sources of antioxidant compounds is the Mediterranean-type diet, especially its content of extra virgin olive oil (EVOO) [17]. Numerous studies have shown the beneficial effects of EVOO and its main antioxidant components (polyphenols) in various aspects of the microangiopathic complications of diabetes mellitus. Most notably, the effects of the main polyphenolic compound of EVOO, hydroxytyrosol (HT), which is a vascular protector and neuroprotector, were demonstrated [11,15,18,19]. However, the effects of HT do not explain all the effects of EVOO, suggesting possible synergy between polyphenols [20]. In an experimental in vitro hypoxia–reoxygenation model in normoglycemic rat brain slices, it was shown that this in vitro synergy occurs to the greatest extent between HT and another EVOO polyphenol, 3′,4′-dihydroxyphenylglycol (DHPG) [21]. We found no studies on the possible neuroprotective effects of DHPG after oral administration; this effect was only described in an in vitro study on the brain tissue of healthy rats [21], so it seemed interesting to determine if this effect could be reproduced in an experimental model of diabetes treated orally with DHPG and HT—alone or in association using the same proportions found in EVOO.

In the different EVOOs of the Mediterranean area, concentrations of HT between 50 and 350 µmol/kg have been described (mean value of all the EVOOs analyzed: 230 µmol/kg), with between 5 and 35 µmol/kg described for DHPG (mean value 21.2 µmol/kg), maintaining a DHPG/HT ratio between 1/5 and 1/10 [22,23].

The aim of this study was to assess the effects of HT and DHPG, alone and in association, in an experimental model of hypoxia–reoxygenation in brain slices of type 1 diabetic rats and on the rate of neuronal cellularity in the retinas of these diabetic animals.

The experimental model of hypoxia–reoxygenation in brain slices of diabetic rats was chosen for a specific purpose: to analyze the neuroprotective effects of DHPG and HT, alone or associated, on nervous tissue, thereby ruling out a possible vascular component in this process. Through in vivo brain ischemia experiments, the final effects of an administered compound could summarize the effects on blood flow and tissue damage mechanisms. On the other hand, analysis of the retinal neurotoxicity produced by diabetes mellitus could provide a way to morphometrically analyze the possible neuroprotective effects of the polyphenolic compounds studied.

## 2. Materials and Methods

### 2.1. Material

The thiobarbituric colorimetric kit for acid-reactive substances and 3-nitrotyrosine enzyme immunoassay kits were obtained from Cell Biolabs Inc. (Bionova Científica S.L., Madrid, Spain). The lactate dehydrogenase colorimetric kit (Cytotoxicity Detection Kit) was obtained from Roche Applied Science, Barcelona, Spain. Total Antioxidant Capacity kits were obtained from Cell Biolabs, Inc. (Arjons Drive, San Diego, CA, USA). Glutathione concentration and glutathione peroxidase activity kits, as well as 8-hydroxy 2 deoxyguanosine kits, were obtained from Abcam plc (Cambridge, CB2 0AX, UK). All other reagents were obtained from Sigma Chemical Corp. (St. Louis, MO, USA).

Hydroxytyrosol was isolated via hydrothermal treatment of the liquid phase obtained from alperujo at 160 °C/60 min [24]. The liquid was extracted using two-step chromatography fractionation. The final yield reached 99.6% purity relative to dry matter, according to the process described by Fernández-Bolaños et al. [25]. The phenols were quantified using a Hewlett-Packard 1100 series equipped with an array diode detector and an Agilent 1100 series automatic injector. A Mediterranean Sea C18 analytical column (250 × 4.6 mm i.d.; particle size = 5 µm) (Teknokroma, Barcelona, Spain) was used at room temperature. The system was equipped with Rheodyne injection valves (20 μL loop). The mobile phases were 0.01% trichloroacetic acid in water (A) and acetonitrile (B), with the following gradient used during the total run time of 55 min: 95% A initially, 75% A at 30 min, 50% A at 45 min, 0% A at 47 min, 75% A at 50 min, and 95% A at 52 min until the run was complete. Quantification was carried out via peak integration at a 280 nm wavelength with reference to calibrations obtained using external standards. Quantification was carried out via peak integration at a 280 nm wavelength with reference to calibrations obtained using external standards.

3′,4′-dihidroxifenilglicol was obtained from the olive oil using a two-phase extraction system employed in olive oil mills to isolate DHPG. The method to purify DHPG was described and patented by Fernández-Bolaños et al. (2010) (WO2010070168A1). The method is based on physical chromatographic systems that allow the extraction of natural compounds without any organic solvent, chemical, or enzymatic reactions, obtaining a purity degree over 95% for dry matter.

### 2.2. Study Design

The animals were 2-month-old adult male Wistar rats (body weight 200–250 g). All rats were used in accordance with current Spanish legislation for animal care, use, and housing (EDL 2013/80847. BOE-A-2013-6271). The recommendations in the Guide for the Care and Use of Laboratory Animals (NIH publication No. 86-23, revised 1985) were followed, as well as the Spanish Law on the Protection of Animals, where applicable. The study protocol was approved by the University of Malaga Ethics Committee for the Use of Animals (Ref. CEUMA31-2018-A) and the Consejería de Agricultura, Ganadería, Pesca y Desarrollo Sostenible, Junta de Andalucía [Department of Agriculture, Livestock, Fisheries and Sustainable Development of the Regional Government of Andalusia] (Ref. 9/07/2019/124).

The animals (10 rats per group) were distributed in seven groups: (1) normoglycemic rats (NDR), (2) diabetic rats (DR) without treatment, except insulin, (3) DR treated with HT (5 mg/kg/day p.o.), (4) DR treated with DHPG 0.5 mg/kg/day p.o., or (5) with 1 mg/kg/day p.o., (6) DR treated with HT plus DHPG 0.5 mg/kg/day p.o., or (7) with HT plus DHPG 1 mg/kg/day p.o.. These doses of each compound were chosen according to the concentrations described in EVOO [21,22]. They ranged from 1/10 to 1/5 (DHPG/HT) depending on the type of EVOO and the effects described in some experiments on healthy and diabetic rats [19,20]. HT and DHPG were administered in the drinking water once daily for 7 days before diabetes was induced and continued daily until the end of the diabetic period (2 months).

Experimental diabetes was induced with streptozotocin (50 mg/kg i.v.). The fasting blood glucose concentration was measured by placing a FreeStyle glucosimeter (Abbot Laboratories S.A., Madrid, Spain) in contact with blood from the saphenous vein. Animals were considered to have diabetes if their blood glucose in fasting condition was higher than 200 mg/dL for 2 consecutive days. Rats in the nondiabetic control group received a single intravenous injection of isotonic saline solution.

During the treatment period, diabetic animals were treated with 4 IU/day s.c. of a soluble long-acting basal insulin analogue (Levemir^®^, Novo Nordisk España, Madrid, Spain) to reduce mortality due to high levels of blood glucose. Control animals received the same volume of isotonic saline solution s.c.

At the end of the second month, all animals from each group were anesthetized with pentobarbital sodium (40 mg/kg i.p.). The brain and both eyeballs were then removed (see below).

#### 2.2.1. Brain Hypoxia–Reoxygenation Procedure

Brain tissue (except for the cerebellum and brain stem) was cut transversally into 1-mm slices with a vibrating microtome (Campden Instruments, San Francisco, CA, USA). The slices were placed in a buffer (composition (mmol/L): 100 NaCl, 0.05 KCl, 24 NaHCO_3_, 0.55 KH_2_PO_4_, 0.005 CaCl_2_, 2 MgSO_4_, 9.8 glucose, pH 7.4) and perfused with a mixture of 95% O_2_ and 5% CO_2_ (Period A). After 30 min, the brain slices were placed in a fresh buffer with no glucose, a CaCl_2_ concentration of 3 mmol/L, and an MgSO_4_ concentration of 0.001 mmol/L. A mixture of 95% N_2_ and 5% CO_2_ was perfused for 20 min (hypoxia) (Period B). Then, the slices were placed in a buffer with glucose and perfused with a mixture of 95% O_2_ and 5% CO_2_ for 120 min (reoxygenation) (Period C).

Two types of samples were analyzed: supernatant samples (before and after the hypoxic period, and after reoxygenation period) and brain samples at the end of the reoxygenation period (frozen immediately in liquid nitrogen and then stored at −80 °C). All analytical techniques were determined within the first seven days after freezing at −80 °C.

#### 2.2.2. Analytical Techniques

All techniques were carried out in a single-blind manner, i.e., the people who did the assays were unaware of the origin and nature of the samples.

##### Lactate Dehydrogenase (LDH) Efflux

LDH efflux was measured in the brain slices supernatant samples as an indirect representation of cell death. Enzyme activity was measured by a colorimetric method according to the manufacturer’s instructions. Briefly, 100 µL of the sample was incubated with 100 µL of a reaction solution (iodotetrazolium and sodium lactate). A blank sample was made with the reaction solution and buffer. The plate was incubated for 30 min, and 50 µL of the stop solution was added. Optical absorbance at 490 and 600 nm was then determined. The value obtained at 600 nm was subtracted from that obtained at 490 nm. Subsequently, the absorbance obtained for the blank was subtracted from all the values.

##### Lipid Peroxidation

Brain tissue was homogenized in 50 mM phosphate-buffered saline, pH 7.0 (1/15 *w*/*v*). Brain homogenized samples were centrifuged at 13,000× *g* for 15 min at 4 °C, freezing the supernatant at −80 °C until the determination of lipid peroxidation.

Thiobarbituric acid reactive substances (TBARS) were measured as an index of lipid peroxides (malondialdehyde (MDA) is the main TBARS compound). Briefly, 100 µL of the sample were incubated for 5 min at 20 °C and then thiobarbituric acid (5.2 mg/mL, pH 3.5) was added. All samples were incubated at 95 °C for 45 min and centrifuged at 3000× *g* for 15 min; the optical absorbance in the supernatant at 532 nm was then measured.

##### 3-Nitrotyrosine

The amount of 3-nitrotyrosine was measured as an index of peroxynitrite formation. Brain slices were homogenized (1:10 wt/vol) in 100 mM KH_2_PO_4_/K_2_HPO_4_ and 0.1% digitonin (pH 7.4). Then, they were centrifuged (5000× *g*, 10 min, 4 °C). The concentration of 3-nitrotyrosine in the supernatant was measured according to the manufacturer’s instructions for the enzyme immunoassay kit. Briefly, 50 µL of samples were incubated with 50 µL of the anti-3-NTy antibody. The plate was incubated with 100 µL of the secondary anti-3-NTy antibody for 1 h at room temperature. The substrate solution was added and the optical absorbance at 450 nm was determined.

##### 8-Hydroxy-2-deoxy-guanosine (8-OHdG)

Brain slices were homogenized 1/15 *w*/*v* in 50 mM phosphate-buffered saline, pH 7.0, and centrifuged at 13,000× *g* for 15 min at 4 °C. The upper layer was freezing at −80 °C until the 8-OHdG concentration in brain slices was determined as an index of oxidative stress DNA damage. An enzyme immunoassay kit was used according to the manufacturer’s protocol.

##### Glutathione Concentration (GSH)

Brain tissue was homogenized in 50 mM phosphate-buffered saline, pH 7.0 (1/25 *w*/*v*), plus 5% sulfosalicylic acid, then was centrifuged at 8000× *g* for 10 min at 4 °C, the upper layer was frozen at −80 °C until the moment of determining the glutathione concentration. A colorimetric assay kit using 5,5′-Dithio-bis-(2-nitrobenzoic Acid) (DTNB) reagent and glutathione reductase was used. Optical absorbance at 412 nm was measured.

##### Glutathione Peroxidase Activity (GSHpx)

Brain tissue was homogenized in 50 mM phosphate-buffered saline, pH 7.0 (1/25 *w*/*v*), then they were centrifuged at 10,000× *g* for 15 min at 4 °C. The upper layer was frozen at −80 °C until the moment of determining glutathione peroxidase activity. The applied colorimetric assay kit was based on the reduction of cumene hydroperoxide by GSHpx while oxidizing reduced glutathione (GSH) to oxidized glutathione (GSSG). In this process, the consumption of NADPH was monitored at 340 nm.

##### Total Antioxidant Capacity (TAC)

Brain tissue was homogenized in 50 mM phosphate-buffered saline, pH 7.0 (1/10 *w*/*v*), then they were centrifuged at 10,000× *g* for 10 min at 4 °C. The upper layer was frozen at −80 °C until the moment of determining the total antioxidant capacity. The assay kit was based on the reduction of copper (II) by antioxidants. The copper (I) ion further reacted with a coupling chromogenic reagent that produced a color with a maximum absorbance at 490 nm.

##### Retina Morphometric Analysis

The eyeballs were removed and fixed in 10% formaldehyde solution for 48 h. Then a conventional protocol of fixation and inclusion in paraffin was carried out and sectioning with an HM 325 rotary microtome (Leica Biosystems, 69226 Nußloch, Germany). Sections were cut at a thickness of 7 μm starting from the posterior area at the level of the optic nerve. Morphometric studies were performed on 40× images. After dewaxing and hematoxylin–eosin staining, quantitative studies of the histological sections were performed in a simple-blind manner by two independent observers to record the thickness between the inner and outer limiting membranes, equivalent to the thickness of the retina. Ganglion layer cells were counted in 40× micrographs after staining, with a total of 10 sections per rat at a known distance from the first section. The results are reported as cell counts per 100 μm segment of retinal length to standardize the data reporting across all samples.

### 2.3. Statistical Analysis

The data in the text, tables, and figures are expressed as the mean ± standard deviation (SD) of ten animals. All statistical analyses were performed with the Statistical Package for Social Sciences v. 25.0 (SPSS Co., Chicago, IL, USA). Unpaired Student’s *t*-tests were used to compare differences between means (control non-diabetic rats vs. control diabetic rats), and an Analysis of Variance test (ANOVA) test was used to establish the differences between treatment groups. To establish a possible relationship between variables, Pearson correlation coefficients were calculated. In all cases, statistical significance was assumed at a value of *p* < 0.05.

## 3. Results

At the end of the follow-up period, the diabetic animals presented lower body weights and ingested greater amounts of food and drink compared to the non-diabetic animals (Table 1). None of the administered compounds significantly modified these parameters. Regarding the serum lipid profile (Table 1), the diabetic animals presented a higher concentration of triglycerides than the normoglycemic animals. HT treatment increased high-density lipoproteins cholesterol (HDL cholesterol) levels and the association of HT with DHPG (0.5 mg/kg/day) and decreased serum triglyceride concentration. Diabetic control animals showed a mean blood glucose concentration higher than that of the non-diabetic control animals after a two-month follow up. These values were not significantly modified in any of the groups of the treated diabetic animals (Table 1).

The experimental hypoxia–reoxygenation model in diabetic rat brain slices produced 75.6% more LDH efflux than that in non-diabetic control animals (Figure 1). HT administration reduced LDH efflux by 42.2% compared to the values of the diabetic control animals. DHPG reduced LDH efflux by 25.5% and 52.5% at 0.5 and 1 mg/kg/day, respectively. The association of HT plus DHPG 0.5 mg/kg/day produced a reduction of 69.8%, while HT plus DHPG 1 mg/kg/day produced a reduction of 71.0% compared to the values of the diabetic control group (Figure 1).

The parameters that define tissue oxidative stress were modified with greater intensity in diabetic animals (Table 2). TBARS, 8-OH-dG, and 3-nitrotyrosine concentrations were, respectively, 2.3, 3.1, and 2.1 times higher compared to those of non-diabetic animals but 58.5% and 32.1% lower for GSH and total antioxidant capacity, respectively, compared to non-diabetic control animals.

HT administration significantly reduced oxidative variables and increased brain antioxidant capacity after hypoxia–reoxygenation (Table 1). DHPG 0.5 mg/kg/day fundamentally reduced the production of TBARS and 8-OH-dG and increased TAC, while 1 mg/kg/day only significantly reduced oxidative variables (Table 1). A statistically significant synergistic effect was observed in terms of TBARS and 8-OH-dG production when HT and DHPG were co-administered (Table 2).

The retinas of the diabetic animals were 80% larger than those of the non-diabetic animals, although this greater thickness was due to an increase in the intercellular space (50.3% greater) since the retinal area occupied by cells was significantly smaller in the diabetic animals (33.2% lower) (Figure 2). The administration of HT or DHPG modified these alterations in the diabetic animals, with a greater effect observed for the association between both polyphenols (Figure 2). The percentage of retinal area occupied by cells decreased by 6.8%, 13%, and 0.8–5% compared to the group of diabetic control animals after administration of HT, DHPG, and HT plus DHPG 1 or 0.5 mg/kg/day, respectively. Practically the same percentages, but in an inhibitory sense, were observed when determining the retinal area occupied by intercellular space.

The retinas of the diabetic control animals presented 68.6% fewer ganglion cells than those in the non-diabetic control group (Figure 3). Diabetic animals treated with HT showed 53.8% fewer ganglion cells compared to non-diabetic animals. Those treated with DHPG 0.5 mg/kg/day showed 63.5% fewer cells, and those treated with DHPG 1 mg/kg/day showed 59.5% fewer cells. When associating HT with DHPG 0.5 mg/kg/day, this reduction was 55.9%, and with HT plus DHPG 1 mg/kg/day, the reduction was 51.8% (Figure 3).

Table 3 shows the statistical correlations between the two cell damage parameters (LDH efflux and percentage of retina occupied by cells) and biochemical variables determined in the brains of the animals after the hypoxia–reoxygenation period. A direct statistical relationship was observed between higher LDH efflux and the variables of oxidative and nitrosative stress; an inverse relationship was observed with the antioxidant variables. The percentage of retinal ganglion cells followed the same pattern and was directly related to LDH efflux.

## 4. Discussion

This study showed that DHPG exerts a neuroprotective effect in a rat model of type 1 diabetes. On the other hand, the association of HT and DHPG in proportions similar to those found in EVOO may increase the effects of both polyphenols separately. This effect is mainly probably due to an antioxidant and anti-nitrosative stress effect in brain tissue subjected to the injured model and a decrease in nitrosative stress at the retinal neuronal level.

A previous study [21] showed that under in vitro conditions, the incubation of HT and DHPG together improved the final neuroprotective effects in a model of hypoxia–reoxygenation in brain slices of normoglycemic rats, the same rats used in this study, thereby relating this effect with antioxidant action and the inhibition of nitrosative stress. One of the main contributions of this study is that we analyzed this interaction between both polyphenols in an experimental model of diabetes in which an increase in brain oxidative and nitrosative stress was previously described. We also observed a greater sensitivity of brain tissue to damage in the hypoxia–reoxygenation model [6]. In this same model, a neurotoxic effect on retinal nerve cells was also demonstrated, with this damage participating in the development of diabetic retinopathy, as well as the long-term vascular damage that develops in diabetes mellitus [15].

Previously published results confirmed the antioxidant and neuroprotective effects of HT in the same experimental model of hypoxia–reoxygenation [19,21,26]. In these studies, the incubation of increasing concentrations of HT in brain slices of healthy rats [21], the oral administration of HT to healthy rats [26], and the application of an experimental model of type 1 diabetes mellitus [19] exerted a neuroprotective effect (LDH efflux) alongside an inhibitory effect on the formation of lipid peroxides (TBARS) and peroxynitrites (3-nitrotyrosine) in brain tissue subjected to the same hypoxia–reoxygenation model. These results agree with those in the present study. Likewise, in ex vivo studies [19,26], a low influence of HT on the glutathione system was observed, as in the present study. The importance of the antioxidant effects of HT on neuroprotective effects, as observed in this study, was also previously established. The use of HT 5 mg/kg/day was based on previous studies using this experimental model [26]. The DHPG doses were chosen based on the composition of the different types of EVOO in a DHPG/HT ratio between 1/10 (5 + 0.5 mg/kg/day) and 1/5 (5 + 1 mg/kg/day) [23].

In terms of the results obtained with DHPG, it was shown that this compound has an antioxidant effect in hydrophilic matrices [27], THP-1 monocytic cell cultures [27], rat liver microsomes [28], and rat brain slices [21]. Likewise, an anti-inflammatory effect was previously demonstrated in THP-1 monocytic cells [27]. This study showed that DHPG exerts cerebral antioxidant effects after oral administration to diabetic animals. These effects are related to DHPG’s neuroprotective effects under the hypoxia–reoxygenation model applied to brain slices of rats, an effect that was already demonstrated under in vitro conditions in normoglycemic animals [21].

When HT and DHPG were associated in the proportion found in the different types of EVOO, the neuroprotective and antioxidant profiles of both polyphenols were improved. This synergistic effect was previously demonstrated through in vitro studies on platelet aggregation [29,30] and in the same experimental model of hypoxia–reoxygenation used in this study in brain tissue from normoglycemic rats [21].

An explanation of this beneficial effect based on associating both polyphenols can be deduced from the statistical correlations found between the two parameters most related to neuronal death in this experimental model and the determined biochemical variables related to oxidative and nitrosative stress (Table 3). We found a clear relationship between the values of LDH (indirect variable of brain cell death), the percentage of retina occupied by cells, and the variables of oxidative and nitrosative stress. This result establishes the fundamental importance of these pathways of cell damage with the loss of living cells in nerve tissue under this experimental model of diabetes. As both polyphenols are clearly antioxidant compounds, we suggest that this effect may be involved in the superior neuroprotective behaviors of both compounds separately [28].

The neuroprotective effects of other bioactive compounds were previously described for experimental models related to ischemia, such as flavonoids, vitamins, and polyphenols [30,31,32]. Most of these models demonstrated a protective effect of brain tissue against ischemic injuries, whether the compounds were administered intravenously (acute phase of experimental stroke) or orally. Several observations were repeated in these studies. On the one hand, the effects of these compounds are related to their ability to inhibit oxidative and nitrosative stress and neuroinflammation; on the other hand, the final results could be a combination of actions at the vascular level and in the nervous tissue itself. Moreover, the possibility of adapting the doses administered in the experimental model to humans was not explored in the majority of previous studies. The present study, by using a hypoxia–reoxygenation model on brain tissue in vitro, ruled out the possible vascular effects of the compounds used. That is, the observed neuroprotective effects were due to direct action in nervous tissue. These results agree with previous studies on the importance of inhibiting oxidative and nitrosative stress in the neuroprotection exerted by HT and DHPG. The translation of these results to humans should also be explored in future studies, the results of which may provide the basis for enriching EVOO to incorporate it into the Mediterranean diet, which was shown to be useful in the prevention of cerebral ischemic events [33].

One limitation of this study relates to the administered doses of these compounds. Most of the studies in which EVOO was administered to analyze its possible beneficial effects in cardiovascular disease recommended the use of 40–50 mL of EVOO per day [33], which implies an intake of approximately 1.5–2.0 mg/day of HT. In this study, we administered 5 mg/kg/day because, although it was not possible to verify the neuroprotective effect on rats with lower doses, an effect on cardiovascular biomarkers was demonstrated [26]. By using polyphenols as molecules extracted from the natural products that contain them, a biological effect is always observed with proportionally higher doses than those contained in these complete products. Similar phenomena occur in the case of HT and DHPG, and the results obtained may serve as a basis for the possible enrichment of EVOO to increase its beneficial effects at the level of nervous tissue in diabetes mellitus.

Another limitation of this study is the lack of analysis of other pathways that could participate in the neuroprotective effects of DHPG in diabetes mellitus. One of our future objectives is to study these effects in an experimental model of type 2 diabetes to analyze any possible influences of both polyphenols on glucose metabolism and thus explain possible additional mechanisms underlying this association in diabetes, as effects were previously described for HT and other natural polyphenols in experimental type 2 diabetes models [34].

Taking into consideration the results obtained in this study, the two EVOO phenolic compounds used do not modify the metabolic profile in the experimental model of type 1 diabetes mellitus, therefore their use could be suggested as an adjunct to the main treatment (insulin, diet, physical exercise) in order to prevent the appearance and development of diabetic neuropathy. In type 2 diabetes, at least hydroxytyrosol has shown a possible hypoglycemic effect, so in this type of diabetes they could be administered, in addition to the prevention of neuropathic alterations, as an aid for the treatment of the metabolic process itself.

## 5. Conclusions

The administration of DHPG or HT to animals subjected to an experimental model of type 1 diabetes mellitus exerted a neuroprotective effect on the brain slices subjected to hypoxia–reoxygenation and on the retina. Their association with the same proportions found in EVOO improved the neuroprotective and antioxidant effects of both polyphenols.

## Figures and Tables

**Figure 1 nutrients-14-01146-f001:**
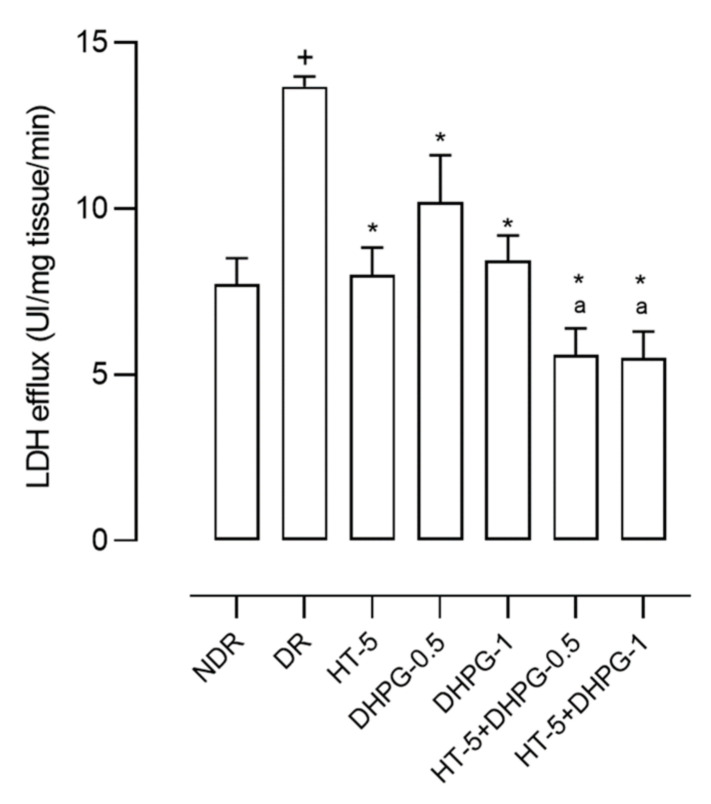
Lactate dehydrogenase efflux (LDH) after the reoxygenation period in brain slices of healthy rats (NDR); two-month follow-up diabetic rats (DR); and DR treated with hydroxytyrosol 5 mg/kg/day (HT-5), 3,4-dihydroxyphenyglycol 0.5 or 1 mg/kg/day (DHPG-0.5 or DHPG-1), HT-5 plus DHPG-0.5 or DHPG-1, p.o. for two months. *n* = 10 rats per group. ^+^
*p* < 0.05 with respect to NDR. * *p* < 0.05 with respect to DR. ^a^
*p* < 0.05 with respect to DHPG-0.5, DHPG-1, and HT-5.

**Figure 2 nutrients-14-01146-f002:**
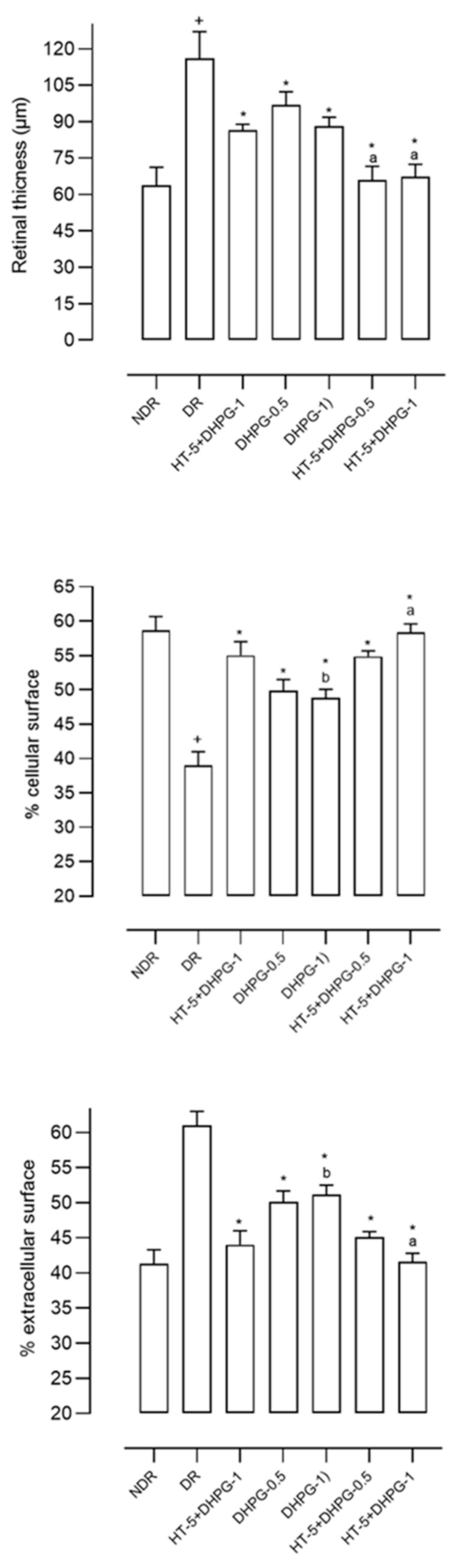
Retinal thickness (upper panel), percentage of retinal area covered by cells (middle panel), and percentage of retinal area covered by extracellular material (lower panel) in retinas from non-diabetic control rats (NDR); two-month follow-up diabetic rats (DR); and DR treated with hydroxytyrosol 5 mg/kg/day (HT-5), 3,4-dihydroxyphenyglycol 0.5 or 1 mg/kg/day (DHPG-0.5 or DHPG-1), HT-5 plus DHPG-0.5, or DHPG-1, p.o. for two months. *n* = 10 rats per group. ^+^
*p* < 0.05 with respect to NDR. * *p* < 0.05 with respect to DR. ^a^
*p* < 0.05 with respect to DHPG-0.5 and DHPG-1. ^b^
*p* < 0.05 with respect to HT-5.

**Figure 3 nutrients-14-01146-f003:**
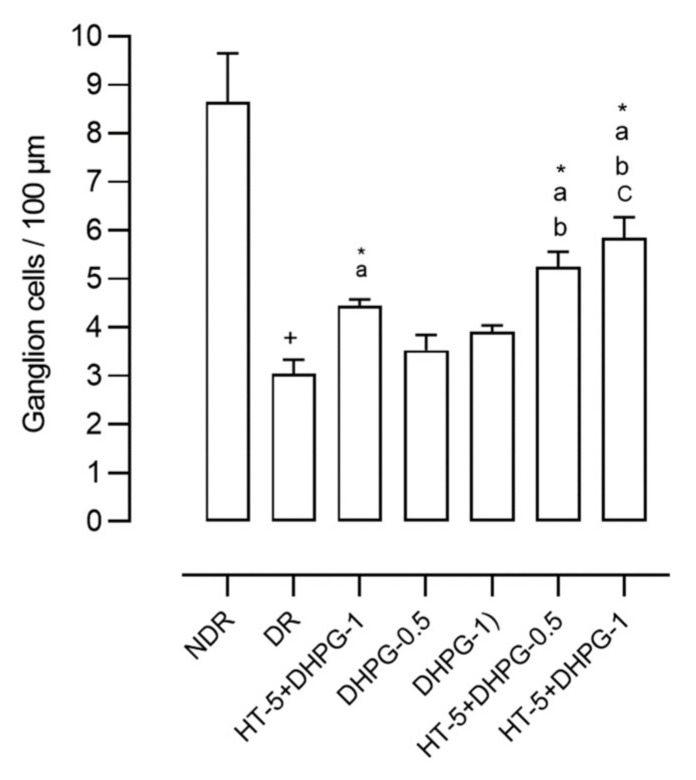
Number of ganglion cells in retinas from healthy rats (NDR); two-month follow-up diabetic rats (DR); and DR treated with hydroxytyrosol 5 mg/kg/day (HT-5), 3,4-dihydroxyphenyglycol 0.5 or 1 mg/kg/day (DHPG-0.5 or DHPG-1), HT-5 plus DHPG-0.5, or DHPG-1, p.o. for two months. *n* = 10 rats per group. ^+^*p* < 0.05 with respect to NDR. * *p* < 0.05 with respect to DR. ^a^
*p* < 0.05 with respect to DHPG-0.5. ^b^
*p* < 0.05 with respect to DHPG-0.5 and DHPG-1. ^c^
*p* < 0.05 with respect to HT-5, DHPG-0.5, and DHPG-1.

**Table 1 nutrients-14-01146-t001:** Zoometric and serum biochemical parameters (mean ± standard deviation) of non-diabetic rats (NDR); diabetic rats (DR); and DR treated with hydroxytyrosol 5 mg/kg/day p.o. (HT-5), 3′,4′-dihidroxifenilglicol (DHPG) 0.5 or 1 mg/kg/day p.o. (DHPG-0.5, DHPG-1), or their associations. N = 10 rats per group.

Parameter	NDR	DR	HT-5	DHPG-0.5	DHPG-1	HT-5+DHPG-0.5	HT-5+DHPG-1
Body weight (g)	448 ± 8.0	340 ± 19.2 ^(+)^	345 ± 17.9	351 ± 20.0	344 ± 15.7	350 ± 22.3	349 ± 21.4
Chow intake (g/day)	20.2 ± 1.8	31.2 ± 3.8 ^(+)^	29.4 ± 2.2	30.8 ± 3.5	29.9 ± 3.0	28.7 ± 2.7	30.0 ± 3.9
Water intake (mL/day)	37.3 ± 4.8	111 ± 9.5 ^(+)^	90.2 ± 15.8	92.1 ± 16.0	95.5 ± 14.6	99.1 ± 16.3	100 ± 19.3
Blood glucose (mg/dL)	98.3 ± 4.8	387 ± 11.2 ^(+)^	370 ± 14.2	369 ± 13.0	377 ± 15.5	380 ± 16.8	381 ± 10.9
Total cholesterol (mg/dL)	60.0 ± 8.1	71.2 ± 5.8	70.0 ± 5.3	75.8 ± 6.4	69.7 ± 8.4	72.3 ± 8.5	71.1 ± 7.8
LDL cholesterol (mg/dL)	22.2 ± 5.0	31.1 ± 6.7	29.8 ± 8.1	30.5 ± 7.0	30.6 ± 7.4	28.9 ± 8.2	28.3 ± 5.9
HDL cholesterol (mg/dL)	19.6 ± 2.9	18.0 ± 5.1	27.6 ± 6.4 ^(^*^)^	23.4 ± 6.6	21.0 ± 5.2	24.5 ± 4.1	21.8 ± 4.8
Triglycerides (mg/dL)	71.4 ± 8.1	110 ± 12.8 ^(+)^	108 ± 11.4	106 ± 12.7	110 ± 14.9	84.4 ± 8.5 ^(^*^,a)^	104 ± 9.8

LDL: low density lipoproteins. HDL: high density lipoproteins. ^+^
*p* < 0.05 with respect to NDR; * *p* < 0.05 with respect to DR; ^a^
*p* < 0.05 with respect to HT-5, DHPG-0.5 and DHPG-1.

**Table 2 nutrients-14-01146-t002:** Oxidative and nitrosative stress parameters (mean ± standard deviation) in brain slices after the hypoxia–reoxygenation period of healthy rats (NDR); diabetic rats (DR); and DR treated with hydroxytyrosol 5 mg/kg/day p.o. (HT-5), 3′,4′-dihidroxifenilglicol 0.5, or 1 mg/kg/day p.o. (DHPG-0.5, DHPG-1) or their associations. *n* = 10 rats per group.

Parameter	NDR	DR	HT-5	DHPG-0.5	DHPG-1	HT-5+DHPG-0.5	HT-5+DHPG-1
TBARS (nmol/mL)	761 ± 106	1746 ± 274 ^+^	492 ± 60.5 *	484 ± 77.8 *	285 ± 58.6 *	165 ± 33.1 *^ac^	126 ± 23.9 *^ac^
8-OHdG (ng/mL)	188 ± 18.7	577 ± 33.8 ^+^	158 ± 17.7 *	117 ± 15.5 *	137 ± 28.9 *^a^	38.3 ± 6.3 *^bc^	24.3 ± 4.4 *^bc^
GHS (nmol/mL)	258 ± 39.9	107 ± 13.2 ^+^	110 ± 7.4	106 ± 13.2 ^c^	129 ± 32.3 ^c^	181 ± 6.0 *	187 ± 4.3 *
GSHpx (nmol/min/mL)	60.8 ± 8.9	87.3 ± 3.4	78.1 ± 6.4	85.0 ± 3.8	72.5 ± 2.4	70.8 ± 4.1	74.5 ± 2.8
TAC (U/mL)	745 ± 22.9	506 ± 52.5 ^+^	732 ± 25.5 *	533 ± 25.7	739 ± 20.2 *	752 ± 24.4 *	745 ± 22.9 *
3-nitrotyrosine (pg/mL)	105 ± 10.2	223 ± 26.4 ^+^	161± 5.8	180 ± 16.1	149 ± 4.8 *	154 ± 8.8 *	149 ± 13.2 *

8-OHdG: 8-hydroxy-2-deoxyguanosine; GSH: reduced glutathione; GSHpx: glutathione peroxidase activity; TAC: total antioxidant capacity; TBARS: thiobarbituric acid-reactive substances. ^+ ^*p* < 0.05 with respect to NDR. * *p* < 0.05 with respect to DR. ^a^
*p* < 0.05 with respect to DHPG-0.5. ^b^
*p* < 0.05 with respect to DHPG-0.5 and DHPG-1. ^c^
*p* < 0.05 with respect to HT-5.

**Table 3 nutrients-14-01146-t003:** Pearson correlations (Pc) between lactate dehydrogenase efflux (LDH), percentage of retinal surface occupied by cells (%CELL), and biochemical parameters in brain slices subjected to the hypoxia–reoxygenation model.

Variable	LDH		%CELL	
	Pc	*p*	Pc	*p*
TBARS	0.848	0.0001	−0.628	0.001
8-OHdG	0.917	0.0001	−0.755	0.0001
GHS	−0.450	0.018	0.580	0.002
GSHpx	0.591	0.002	−0.386	n.s.
TAC	−0.829	0.0001	0.609	0.001
3-Nty	0.730	0.0001	−0.615	0.001
LDH	-----	-----	−0.709	0.0001

n.s.: non-significant; Pc: Pearson coefficient; *p*: *p* value; TBARS: thiobarbituric acid reactive substances; 8-OHdG: 8-hydroxy-2-oxyguanosine; GSH: reduced glutathione; GSHpx: glutathione peroxidase activity; TAC: total antioxidant capacity; 3-Nty: 3-nitrotyrosine.

## Data Availability

The data presented in this study are available in the article.

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
