# Peer review of "Neuroprotective Effect of 3′,4′-Dihydroxyphenylglycol in Type-1-like Diabetic Rats—Influence of the Hydroxytyrosol/3′,4′-dihydroxyphenylglycol Ratio"

_nutrients, 2022, doi:10.3390/nu14061146_

Round 1
Reviewer 1 Report
This study describe the neuroprotective effect of HT and DHPG bioactive compounds which are present in EVOO.
Mayor concern
Abstract needs to be revised. The English is very poor, including sentences without verb and lack of coherence and accuracy.
The doses of each compound were chosen according to the range of concentrations described in EVOO with different polyphenol content. Please, explain and discuss in the text if these dose are achievable within the context of a balance diet. Additionally, reported concentration of hydroxytyrosol and 3',4'-dihydroxyphenylglycol in olive oil (any kind) should be included in the introduction.
Lines 49-51. This paragraph should be broaden; oxidative stress is very general. Please, explain which oxidative stress biomarkers are associated to microangiopathic processes, especially those related to the present study.
Lines 273-275 and 294-296: Please describe the results of these analysis
Lines 321-323: Please, compare and discuss these previously publish results whit the present results.
Discussion section: Discussion and comparison with the reported effect of other bioactive compounds should be included
I do not agree with the conclusions. HT alone also exert neuroprotective effect. Please rephrase it.
Minor concern
Line 82. It meant to be HP Agilent 1100?
Lines 86-88: Please, indicate the names of mobiles phases (A or B)
Line 173: 3-Nitrotyrosine
Line 185: 8-Hydroxy-2-deoxy-guanosine
Lines 323-326: Please, rephrase this sentence. It is too long and hard to read
Author Response
Reviewer 1
Abstract needs to be revised. The English is very poor, including sentences without verb and lack of coherence and accuracy.
Response: Abstract section has been rewritten in this revised version. Moreover, the revised manuscript has been sent to English Editing Services (MDPI).
The doses of each compound were chosen according to the range of concentrations described in EVOO with different polyphenol content. Please, explain and discuss in the text if these doses are achievable within the context of a balance diet. Additionally, reported concentration of hydroxytyrosol and 3',4'-dihydroxyphenylglycol in olive oil (any kind) should be included in the introduction.
Response: A paragraph in the introduction, just before the aim of the study, has been added the range of concentrations of HT and DHPG described in EVOO with different polyphenol content. Moreover, at the end of discussion we have been added a paragraph in which the dose of the polyphenol compounds is discussed as a limitation of the study.
Lines 49-51. This paragraph should be broadened; oxidative stress is very general. Please, explain which oxidative stress biomarkers are associated to microangiopathic processes, especially those related to the present study.
Response: In this revised version we have included an explanation about oxidative stress in diabetes, including new references.
Lines 273-275 and 294-296: Please describe the results of these analysis
Response: These results have been included in this revised version.
Lines 321-323: Please, compare and discuss these previously publish results whit the present results.
Response: These results has been compared with those of the present study.
Discussion section: Discussion and comparison with the reported effect of other bioactive compounds should be included
Response: In the revised manuscript, a new paragraph (just before the limitations of the study) has been added including the effect of other bioactive compounds.
I do not agree with the conclusions. HT alone also exert neuroprotective effect. Please rephrase it.
Response: HT has been included in conclusions.
Minor concern
Line 82. It meant to be HP Agilent 1100?
Response: Hewlett-Packard 1100 series equipped with an array diode detector and an Agilent 1100 series automatic injector.
This sentence has been included.
Lines 86-88: Please, indicate the names of mobiles phases (A or B)
Response: The mobile phases were 0.01% trichloroacetic acid in water (A) and acetonitrile (B), with the following gradient during a total run time of 55 min: 95% A initially, 75% A at 30 min, 50% A at 45 min, 0% A at 47 min, 75% A at 50 min, and 95% A at 52 min until the run was complete. Quantification was carried out by peak integration at 280 nm wavelength with reference to calibrations obtained with external standards.
This sentence has been included.
Line 173: 3-Nitrotyrosine
Response: It has been corrected.
Line 185: 8-Hydroxy-2-deoxy-guanosine
Response: It has been corrected.
Lines 323-326: Please, rephrase this sentence. It is too long and hard to read
Response: This semtence has been simplified.
Reviewer 2 Report
In this work, Rodriguez-Pérez and colleagues intended to explore the neuroprotective potentiality of two flavonoids (Hydroxytyrosol (HT) and 3',4'-dihydroxyphenylglycol (DHPG)) present in extra virgin olive oil in an animal model of type 1 diabetes. Even though the work shows some interesting data, the manuscript requires a revision before being considered for publication. Please take into consideration the following comments/suggestions when revising the manuscript in order to improve its significance.
1- There are some grammar incongruences throughout the manuscript. It should be revised by someone fluent in English.
2- The introduction section needs to be improved. The authors present some literature and their work hypothesis but there is no explanation why the work was performed in brain slices from diabetic animals exposed to a hypoxia-reoxygenation insult.
3- The same way, it is not clear why the retinas of the animals were used in the experiments.
4- Regarding the experimental groups, there should be a diabetic group alone, i.e. without hypoxia-reoxygenation. This would be important to present and explain the study outcomes. Again, why this insult was used is not clear and it should be well explicated.
5- The manuscript must have a table with animals’ characterization at time of sacrifice. It must contain information on animals´ weight and glycaemia. Biochemical parameters like cholesterol and triglycerides would be a plus to understand if the flavonoids exert any systemic effect. What about water consumption? Did the flavonoids change the water intake throughout the treatment?
6- In table 1 and 2, the word “variable” to designate the parameter analyzed is not accurate. Please change it.
7- The authors focused their work in evaluating the oxidative status of the brain slices but did not went further and investigated the functional and histological outcomes of the treatment. This is a significant limitation of the study. Please consider to perform those additional experiments before trying another submission.
Author Response
Reviewer 2
1- There are some grammar incongruences throughout the manuscript. It should be revised by someone fluent in English.
Response: The revised manuscript has been sent to English Editing Services (MDPI).
2- The introduction section needs to be improved. The authors present some literature and their work hypothesis but there is no explanation why the work was performed in brain slices from diabetic animals exposed to a hypoxia-reoxygenation insult.
Response: In this revised version we have included a last paragraph in the introduction to explain why we analyzed the hypoxia-reoxigenation model and the retinal analysis.
3- The same way, it is not clear why the retinas of the animals were used in the experiments.
Response: See above.
4- Regarding the experimental groups, there should be a diabetic group alone, i.e. without hypoxia-reoxygenation. This would be important to present and explain the study outcomes. Again, why this insult was used is not clear and it should be well explicated.
Response: Thank you very much for your comment. It is true, we have not included groups without the hypoxia-reoxygenation model. In the literature, bioactive compounds with antioxidant power have been described as having little effect on health or basal conditions, while a clear effect is seen when inducing oxidative damage. Same goes for HT. That is why we do not consider assessing the effect of these compounds without using a cerebral oxidative injury.
5- The manuscript must have a table with animals’ characterization at time of sacrifice. It must contain information on animals´ weight and glycaemia. Biochemical parameters like cholesterol and triglycerides would be a plus to understand if the flavonoids exert any systemic effect. What about water consumption? Did the flavonoids change the water intake throughout the treatment?
Response: A new Table 1 has been included containing zoometric and serum lipid parameters at the end of follow-up.
6- In table 1 and 2, the word “variable” to designate the parameter analyzed is not accurate. Please change it.
Response: It has been changed.
7- The authors focused their work in evaluating the oxidative status of the brain slices but did not went further and investigated the functional and histological outcomes of the treatment. This is a significant limitation of the study. Please consider to perform those additional experiments before trying another submission.
Response: Thank you very much for your comment. Indeed, the next step to advance in this line of research is to verify that the findings described at the biochemical and morphometric level correspond to a functional modification, which is possibly also affected in diabetes mellitus. The objective of our study was to describe these biochemical changes in the experimental model used, results that had not been published with DHPG alone or associated with HT. Undoubtedly, the next works in this line will include functional experiments.
Round 2
Reviewer 2 Report
Overall, the authors tried to satisfactorily address all the raised comments and suggestions.
1- There are still some incongruences and typos. For instance, in the results section: “Non-diabetic control animals showed a mean blood glucose concentration higher than that of the diabetic control animals …”. I believe that the authors were trying to say that the control animals presented a lower blood glucose levels than the diabetic animals.
2- Please elucidate if the blood glucose levels were taken under a fasting or fasted condition.
3- Regarding the experimental groups, the authors present some reasons to not have included a diabetic group treated with the flavonoids. In detail, the authors say “In the literature, bioactive compounds with antioxidant power have been described as having little effect on health or basal conditions, while a clear effect is seen when inducing oxidative damage”. However, and please let me to disagree with you, diabetes per se cannot be considered a healthy or basal condition; diabetes per se is known to induce disturbances in the oxidative stress machinery in the brain tissue as well as in the retina. The question arising from here is, would those compounds exert the same neuroprotective effect if there was no extra insult? I do understand that having an extra group of animals can be costly and laborious but, please take this into consideration when designing further studies.
4- Considering that none of the treatments modified the diabetic systemic phenotype, what is the clinical importance of taking supplementation with the phenolic compounds under a diabetic context? Can the authors speculate on this? Suggest a combination therapy?
5- The word “variable” to designate the parameter analyzed continues to appear in table 1 and table 2. Please change this.
Author Response
RESPONSE TO THE REVIEWER #2
Overall, the authors tried to satisfactorily address all the raised comments and suggestions.
Response: Thank you very much.
1- There are still some incongruences and typos. For instance, in the results section: “Non-diabetic control animals showed a mean blood glucose concentration higher than that of the diabetic control animals …”. I believe that the authors were trying to say that the control animals presented a lower blood glucose levels than the diabetic animals.
Response: Thank you for your comment. This sentence has been changed in the correct form.
2- Please elucidate if the blood glucose levels were taken under a fasting or fasted condition.
Response: Blood glucose was measured in fasting condition. This has been indicated in line 154.
3- Regarding the experimental groups, the authors present some reasons to not have included a diabetic group treated with the flavonoids. In detail, the authors say “In the literature, bioactive compounds with antioxidant power have been described as having little effect on health or basal conditions, while a clear effect is seen when inducing oxidative damage”. However, and please let me to disagree with you, diabetes per se cannot be considered a healthy or basal condition; diabetes per se is known to induce disturbances in the oxidative stress machinery in the brain tissue as well as in the retina. The question arising from here is, would those compounds exert the same neuroprotective effect if there was no extra insult? I do understand that having an extra group of animals can be costly and laborious but, please take this into consideration when designing further studies.
Response: Thank you very much for your comment. We completely agree with your comment. Diabetes itself induces tissue changes (oxidative and nitrosative stress, etc.). As we indicated in the discussion of the manuscript, an antioxidant and neuroprotective effect of hydroxytyrosol and DHPG has been demonstrated (our group has published several papers on the matter) in the same experimental diabetes model used in this work, measuring serum cardiovascular biomarkers. We will take your suggestion into account with future studies regarding nervous tissue in diabetes.
4- Considering that none of the treatments modified the diabetic systemic phenotype, what is the clinical importance of taking supplementation with the phenolic compounds under a diabetic context? Can the authors speculate on this? Suggest a combination therapy?
Response: In this second revised version of the manuscript, a new paragraph, just before the conclusions, has been added. We suggest the use of these compounds to prevent neuropathic damage in type 1 diabetes mellitus, and to help the treatment of hyperglycemia in type 2 diabetes.
5- The word “variable” to designate the parameter analyzed continues to appear in table 1 and table 2. Please change this.
Response: This has been changed in this version.